# Pioglitazone as a Possible Treatment for *Ataxia-Telangiectasia*

**DOI:** 10.3390/biom14101264

**Published:** 2024-10-08

**Authors:** Rodney Shackelford

**Affiliations:** Department of Pathology, University of South Alabama, 2451 University Hospital Drive, Mobile, AL 36617, USA; rshackelford@health.southalabama.edu or rdnyshac@aol.com; Tel.: +318-525-6479

**Keywords:** *ataxia-telangiectasia*, pioglitazone, glutathione, iron–sulfur complex

## Abstract

*Ataxia-telangiectasia* (AT) is a rare autosomal recessive disorder characterized by immunodeficiency, progressive cerebellar ataxia, and an increased malignancy risk. Cells derived from individuals with AT show multiple defects, including high oxidant and ionizing radiation sensitivities, poor DNA repair, low iron–sulfur cluster levels, and low reduced glutathione. The clinical course of AT is progressive and unrelenting, with most individuals having a survival time of approximately twenty-five years. Presently, AT has no effective treatments, and most patients receive supportive care only. Recently, pioglitazone, a thiazolidinedione class used to treat type 2 diabetes, has been demonstrated to exert beneficial effects on AT cells and on diabetic individuals with AT. Here, I will discuss the possible molecular mechanisms of pioglitazone’s favorable effects on the AT phenotype and why it may have utility in treating some aspects of AT.

## 1. Introduction

### 1.1. Ataxia-Telangiectasia

*Ataxia-telangiectasia* (*AT*) AT is a rare autosomal recessive disorder secondary to biallelic mutations of the *ataxia-telangiectasia mutated* (*ATM*) gene [1,2]. AT is a multi-system disease with an unrelenting and progressing clinical course and an average survival time of 19 to 25 years [1,2,3,4,5]. The estimated incidence of AT is roughly 1 out of 88,000 live births in the USA (1 in 40,000–300,000 live births) with the onset of symptoms usually appearing between 1 and 5 years [1,2,3,4]. AT patients show a wide range of clinical manifestations, with one study of 1314 individuals with AT presenting with 2134 symptoms [4]. Presently there are 2514 globally reported patients with AT; however, the number of afflicted individuals is almost certainly much higher, due to AT being significantly underdiagnosed [1,2,3,4,5]. Presently, AT has no curative treatment [1,2,3,4,5].

The most common symptoms of AT include neurological signs, such as progressive cerebellar ataxia, chorea, choreoathetosis, nystagmus, saccadic intrusions, dystonia, dysphagia, and oculomotor apraxia [4,6,7]. The progressive ataxia generally presents at 12–18 months, secondary to cerebellar Purkinje and granular cell death, which gradually worsens, leaving most patients wheelchair bound by age 10 [4,6,7]. Other symptoms of AT are oculocutaneous telangiectasias, usually appearing by ages 3–4, that are easily identified on the bulbar conjunctiva, and commonly occur on sun-exposed areas and other organs such as the brain and bladder [4,8,9]. Immunodeficiency is common in AT, with over 60% of individuals showing low IgA, IgE, and IgG levels, variable IgM, and, often, reduced B and T lymphocytes [1,2,3,4,5,10]. This immunodeficiency confers a predisposition to recurrent infections, especially sinopulmonary infections and chronic pulmonary inflammatory disease, which over time can evolve into interstitial lung disease and bronchiectasis [10,11].

AT patients also show a 25–40% lifetime cancer risk, with leukemias and lymphomas being common, especially T-cell acute lymphoblastic and T-cell prolymphocytic leukemias, which account for 85% of AT childhood malignancies [12,13]. Later in life, AT patients remain susceptible to lymphoid tumors and become subject to different solid tissue tumors, such as gastric, liver, breast, parotid, and esophageal carcinomas [1,2,3,4,5,12,13]. Other abnormalities seen in AT include insulin-dependent diabetes, progeric changes, growth failure, gonadal atrophy, and cardiovascular disease [1,2,3,4,5]. The two most common causes of death in AT are malignancy and chronic pulmonary disease [1,2,3,4,5,10,11,12,13]. Interestingly, a hallmark of AT is elevated serum α-fetoprotein, which increases with age and has been linked to the neurodegeneration in AT [1,2,3,4,5,14].

### 1.2. ATM and AT

The gene mutated in AT, *ataxia-telangiectasia mutated* (*ATM*), is localized to chromosome 11q22.3–23.1, contains 66 exons, and encodes a 350 kDa serine/threonine kinase which phosphorylates threonine or serine residues, then phosphorylates a glutamine [1,2,3,4,5]. *ATM* belongs to the phosphatidylinositol 3-kinase-related protein kinase family which includes *ATM*, the mammalian target of rapamycin, a suppressor of morphogenesis in genitalia, DNA-dependent protein kinase, transformation/transcription domain-associated protein, and the *ATM* and Rad-3-related protein kinase [1,2,3,4,5,15,16,17]. ATM is activated by many different stimuli, with ionizing radiation, oxidative stress, and double-stranded DNA beaks being the most characterized [15,16,17]. ATM phosphorylates over 1100 different targets and regulates a vast number of different cellular events, including cell cycle checkpoints, homologous recombination, DNA repair and genomic stability maintenance, the cellular nutrient sensing status, oxidative stress resistance, telomere and stem cell maintenance, mitochondrial function, insulin signaling, glutathione (GSH) biosynthesis, and iron–sulfur (Fe–S) cluster formation [1,2,3,4,5,18,19,20].

More than 1400 ATM mutations have been identified in AT with most being nonfunctional truncating mutations. Less commonly, missense and splicing variant mutations occur, often resulting in a milder AT phenotype [1,2,3,4,5]. Cells from individuals with AT show multiple deficits compared to wild-type cells, including high sensitivity to oxidants, ionizing radiation, and labile iron. Additionally, poor and attenuated double-stranded DNA break repair is seen, resulting in chromosomal instability. Also, following ionizing radiation exposure, AT cells also show slow p53 and cell cycle checkpoint induction. Lastly, increased mitochondrial reactive oxygen species (ROS) formation and changes in the cellular redox status are seen, including altered glutathione redox status and low catalase activity [1,2,3,4,5,15,16,17,18,19,20,21,22,23,24]. Together, these deficits result in a premature aging phenotype with dysfunctions in many systems. For example, the poor ability to repair DNA results in poor immunoglobin synthesis, contributing to impaired immunity, while the high cancer risk comes from high genomic instability. The neurodegeneration seen in AT is poorly understood, but dysregulated antioxidant defenses, especially of cystine import and GSH synthesis, likely lead to neural cell death [1,2,3,4,5,15,16,17,18,19,20,21,22,23,24].

### 1.3. AT and Dysregulated Iron Metabolism

Some of the pathology seen in AT appears to be related to aberrant iron metabolism. For example, *Atm*-deficient mice also show increased serum and hepatic iron, with higher ferritin and hepcidin levels [25]. Additionally, in culture, AT cells exhibit an unusual sensitivity to labile iron and show increased growth and genomic stability in the presence of iron chelators or iron-free media, events not seen in ATM wild-type cells, or in AT cells corrected with recombinant ATM protein expression [22,23,24]. The reason for the increased AT cell labile iron sensitivity likely involves iron-catalyzed ROS generation and the poor ability of AT cell to repair double-stranded DNA breaks [1,2,3,4,5,16,17,18,22,23,24,25]. Recently, an analysis of cultured AT cells with and without recombinant ATM protein expression revealed that ATM expression in AT cells increased clusters levels roughly five-fold and interestingly, also significantly increased the low AT cell expression of the NFS1 and NFU1 proteins. NFS1 plays a central role in Fe–S cluster synthesis, as it catalyzes the first step in cluster synthesis. NFU1 is required to make more complex Fe–S clusters containing three and four iron atoms. Without these proteins, iron metabolism is severely disrupted [19]. Additionally, recombinant ATM expression in cultured AT cells significantly increases the cellular reduced to oxidized GSH ratio roughly 10-fold (GSH: GSSG ratio) [19].

## 2. AMPK-α Phosphorylation, ROS, Pioglitazone, and AT

### 2.1. AMPK-α Phosphorylation

The molecular mechanisms by which the ATM kinase increases iron sulfur cluster formation and increases the GSH: GSSG ratio are unknown. However, ATM phosphorylates and activates the adenosine monophosphate-activated protein kinase-α (AMPK-α), which phosphorylates and stabilizes the iron sulfur cluster assembly enzyme (ISCU) protein complex, enhancing iron sulfur cluster synthesis, maintaining complex function, and also phosphorylating glutathione reductase on threonine 507, increasing the enzyme’s activity and increasing cellular reduced glutathione [26,27]. AMPK-α activation requires the phosphorylation of threonine 172 (T172) in the AMPK-α activation loop [28,29]. AMPK-α T172 is phosphorylated by ATM, LKB1, Ca^2+^/calmodulin-dependent protein kinase β, and possibly TAK1 kinase [28]. Multiple studies employing tissue-specific LKB1 deletions indicated that most AMPK-α activation is due to LKB1 activity [28]. Different studies have shown different results on the effects of ATM and LKB1 in phosphorylating AMPK-α [30,31,32]. In one study, inducible AMPK-α T172 phosphorylation in ATM null cells was attenuated, but still occurred, while no phosphorylation was seen in the LKB1 null cells [30]. Other studies revealed that inducible AMPK-α phosphorylation requires ATM and occurs without LKB1 expression [31,32]. Thus, it appears ATM and LKB1 activates and phosphorylates AMPK-α with different patterns in different cell types, but ATM clearly plays a role in induced AMPK-α phosphorylation [30,31,32].

Interestingly, unstimulated AMPK-α phosphorylation is not low in AT cells, but is increased compared to wild-type cells [32,33]. In one study, steady-state AMPK-α phosphorylation in the cerebelli of *Atm*-deficient mice was significantly increased compared to syngeneic wild-type mice. The *Atm*-deficient mice showed high ROS and malondialdehyde levels, accompanied by neurodegeneration and neuromotor deficits. The treatment of these mice with monosodium luminol lowered their cerebellar astrocyte AMPK-α phosphorylation and malondialdehyde levels, while improving their murine neuromotor performance [33]. Based on this, the paper’s authors hypothesized that lowering AMPK-α activation could have value in treating AT, particularly in its neurodegeneration [33]. The molecular mechanism underlying the increased AMPK-α phosphorylation in AT is unknown; however, there are two likely reasons for this event. First, AT cells show deficient E3-ubiquitin ligase activity, resulting is lowered target protein ubiquitination and degradation [34]. The LKB1 kinase expression and activity are post-transcriptionally regulated, in part, by E3-ubiquitin ligase activity [35]. Thus, in AT cells where E3-ubiquitin ligase activity is diminished, there should be increased LKB1 protein and activity and subsequent AMPK-α phosphorylation. As LKB1 mediates the majority of AMPK-α phosphorylation, changes in its expression should have a significant impact on AMPK-α phosphorylation [28]. Additionally, AT is charactered by increased cellular oxidative stress, with increased NOX4 levels and mitochondrial dysfunction being major ROS sources [36,37]. Both the LKB1 and CaMKKβ kinases are activated by oxidative stress, leading to increased AMPK-α phosphorylation and activity; thus the chronic oxidative seen in AT may also contribute to AMPK-α phosphorylation [36,37,38]. Support for this comes from the observation that increased AMPK-α phosphorylation in AT is attenuated by antioxidant treatment [33].

### 2.2. Pioglitazone, AMPK-α T172 Phosphorylation, and AT Cells

Based on the data above, a pharmacologic agent that modulates AMPK-α phosphorylation and has antioxidant properties may exert beneficial effects on AT cells and AT patients. The thiazolidinedione class drug pioglitazone increases T172 AMPK-α phosphorylation in cultured rat insulinoma cells and also exerts powerful antioxidant properties, increasing reduced GSH, while lowering MDA, TNF-α, IL-6, and ROS levels in various studies [39,40,41,42,43,44,45]. Additionally, pioglitazone increases glutathione reductase activity and mitochondrial complex I activity, both of which are deficient in AT [20,41,42,43,44,46]. The mechanisms by which pioglitazone induces AMPK-α phosphorylation are unknown, although LKB1 activation by events such as adipocyte adiponectin release and AMP accumulation are likely [28,47]. Pioglitazone is an FDA-approved drug used to treat type 2 diabetes in adults and targets insulin resistance by activating the peroxisome proliferator-activated receptor-γ. It is generally administered orally once per day [48].

When an AT cell line and the same cell line with recombinant ATM protein expression were cultured with pioglitazone, the AT cells showed (1) a significantly increased iron sulfur cluster fraction (~five-fold), (2) a significantly increased GSH: GSSG ratio (~10-fold), and (3) significantly lowered oxidative stress-induced double-stranded DNA beak formation [49]. These effects were not seen in the recombinant ATM expression-corrected AT cell line [49]. These results demonstrate that pioglitazone can exert beneficial effects on AT cells that are not seen in ATM-corrected cells. Interestingly, pioglitazone significantly induced NFS1 mRNA in both AT cells and recombinant ATM protein-corrected AT cells, demonstrating the novel effect of this drug on the initial step in the iron cluster synthesis pathway [49]. The in vitro pioglitazone concentrations used in these experiments was 30 μM [49]. Three μM pioglitazone also significantly protected AT cells from the damaging effects of an exogenous oxidant in the colony–efficiency forming assay [49]. In other studies, 10 μM in vitro pioglitazone increased the GSH: GSSG ratio in cultured pancreatic cells, and increased GSH reductase activity and GSH redox cycling under different conditions of stress [39,40,41,42,43,44]. Specifically, pioglitazone improved cellular function in cultured cells and animals stressed by different events, including high glucose, cisplatin exposure, and the pathophysiology present in an X-linked adrenoleukodystrophy animal model and in cultured AT cells [39,40,41,42,43,49]. Thus, it appears that pioglitazone promotes the survival of stressed cells, including AT cells [49]. Lastly, the reported blood concentrations of pioglitazone are 3–6 μM, indicating that the concentrations used in the AT cell culture are near physiologically effective concentrations [49,50].

### 2.3. Dysregulated GSH and Iron Metabolism in AT and the Effects of Exogeneous Thiol Supplementation on AT Cells

Part of this hypothesis is that increasing cellular-reduced GSH might benefit AT patients. There are numerous studies indicating that this is possible, as AT cells show aberrant GSH and sulfur metabolism that is corrected by changes in sulfur and GSH metabolism [20,51,52]. First, the red blood cells from AT patients show lower GSH levels, while fibroblasts and lymphocytes from AT patients show slow and impaired GSH synthesis compared to wild type individuals, accompanied by very slow cystine import. The GSH synthesis defect was corrected when the cells were permeabilized, allowing adequate cystine import, showing that increasing GSH precursor bioavailability corrected this AT cell defect [51,52]. In a short-term study, AT patients were given betamethasone, resulting in increased intracellular GSH that was accompanied by an improved Scale for the Assessment and Rating of Ataxia (SARA) score. The greatest improvements were seen in the AT patients who showed the highest GSH levels [53].

Exogenous thiol-based antioxidants have shown utility in attenuating AT-related deficiencies in *Atm*-deficient mice, AT cells, and in an AT patient. For example, in *Atm*-deficient mice, dietary N-acetyl cysteine lowered the 8-OH deoxyguanosine levels, the DNA deletion frequency, and the incidence of multiplicity of murine lymphomas, and increased the murine lifespan [54,55]. Similarly, exogenous α-lipoic acid given to AT cells reduced the ROS levels and mitochondrial dysfunction [56]. Additionally, the astroglia from these mice showed low cystine/glutamate exchanger subunit xCT, low GSH-S-transferase expression, and low intracellular and excreted GSH. These cells showed impaired L-cystine import, the rate-limiting precursor for GSH synthesis, which impaired cell survival. Circumventing the xCT-dependent import of L-cystine through N-acetyl-L-cysteine supplementation restored GSH levels in the astroglia and increased cell survival [20]. Lastly, in one case study, a nine-year-old AT patient was given N-acetyl-DL-leucine at 4 g/day for 16 weeks, resulting in significantly improved ataxia symptoms and quality of life [57]. The utility of this drug in treating AT is currently being investigated [58].

The other major effect of pioglitazone on AT cells, increasing Fe–S cluster levels (low in AT cells), may benefit AT cells, as it is a significant correction of one of the deficits seen in AT cells. It remains to be determined if the pioglitazone-induced increased Fe–S cluster will improve AT mitochondrial function and other Fe–S cluster-dependent cellular events. The reduction in oxidative stress-induced double stranded DNA breaks and increased cell viability following oxidative stress seen in pioglitazone-treated AT cells are likely due to an improved cellular redox status [49]. Taken together, the above data indicates that reduced thiol compounds and an increase in reduced GSH have beneficial effects on AT cells, *Atm*-deficient mice, and AT patients. Thus, the ability of pioglitazone to activate GSH reductase, increase reduced GSH levels, and raise the GSH: GSSG ratio, indicates that it may have utility in treating AT. A hypothetical model of the molecular pathways of pioglitazone’s effects on AT cells is shown in Figure 1.

Other AMPK-α activators exist, with metformin being prominent [59]. However, AMPK-α activation by metformin only occurs at supra-pharmacological (>1 mM) concentrations, which do not occur in the clinical setting [59]. Other data indicates that metformin does not act through AMPK-α. For example, Foretz et al. [60], found that hepatocytes with LKB1 knockout were metformin responsive and a liver-specific deletion of AMPK also did not suppress metformin action. Based on current data, it appears that pioglitazone can activate AMPK-α at 10 μM concentrations, significantly lower than metformin [39,59].

### 2.4. Studies on Pioglitazone Administration to Diabetic AT Patients

Support for this hypothesis comes from a recent study wherein eight nondiabetic people with AT were compared to 15 healthy controls matched for age and HbA1c levels, all of whom were treated with pioglitazone. Pioglitazone treatment increased insulin sensitivity only in the AT group and improved adipocyte function both groups. Additionally, pioglitazone reduced plasma non-esterified fatty acids and increased the fasting respiratory quotient, indicating lowered fatty acid oxidation in the AT patients. Interestingly, this study employed a mixed effects model of insulin sensitivity to identify a significant gene–drug interaction between AT and pioglitazone. The study concluded that pioglitazone may be an appropriate first-line treatment for diabetes in AT patients [61].

### 2.5. Low Thiol-Based Antioxidant Compounds, AT, NFS1, and NFU1

NFU1 protein levels are extremely low in AT cells by mass spectrometric analysis compared to the ATM protein expression corrected cells [19]. NFU1 functions in the assembly of higher order Fe–S clusters (4Fe-4S) and in the regulation lipoic acid synthetase [49,62]. NFU1 mutations result in a lipoic acid synthetase deficiency and low cellular lipoic acid concentrations [62]. Thus, while NFU1 in not mutated in AT, the extremely low levels of this protein in AT suggests that lipoic acid synthetase activity and cellular lipoic acid concretions are low in AT cells due to an NFU1 protein deficiency, contributing to the AT cell redox pathology [51,52,53,54,55,56,57]. This may, in part, explain why exogenous thiol-based antioxidants exert beneficial effects on AT cells.

### 2.6. Pioglitazone and Its Possible Side-Effects

Pioglitazone is FDA approved to treat type 2 diabetes in adults and targets insulin resistance by activating the peroxisome proliferator-activated receptor-γ [48]. It has also been successfully used to treat diabetes in individuals with AT [61]. The drug is not generally used in the pediatric population [48]. The main side effect of pioglitazone has been a possible association with bladder cancer. The risk of bladder cancer appears to be very low, with a meta-analysis published in 2017 showing no link between long-term pioglitazone use and bladder cancer [63]. However, other meta-analyses have demonstrated an association, which appears to be mild and increased with higher pioglitazone doses and long-term use [64,65,66]. Most studies have concluded that the use of pioglitazone is justified, and the bladder cancer risk is very low [48,63,64,65,66]. However, as individuals with AT show an elevated cancer risk and are usually within the pediatric population, this possible effect of pioglitazone needs to be considered [1,2,3,4,5]. Since, in culture, pioglitazone exerts beneficial effect of AT cells at lower 3 μM concentrations, a lower dose of the drug may be efficacious and lower any bladder cancer risks [43].

The neurodegeneration in AT is one is one of the main causes of the disease’s morbidity and mortality [1,2,3,4,5]. Pioglitazone usually does not cross the blood-brain barrier, although there are some formulations that allow this. There is evidence, however, that the drug exerts beneficial effect in the brain, where it attenuates oxidative stress, and enhances neurogenesis, synaptic plasticity, and mitochondrial function [45]. These aspects of pioglitazone’s function may also help treat the neurodegeneration in AT.

## 3. Conclusions: Pioglitazone as a Possible Treatment for AT

The above data suggests that pioglitazone may have efficacy in treating AT, particularly as it increases the GSH: GSSG ratio and the Fe–S clusters specifically in AT cells and has shown beneficial effects in AT patients with diabetes [49,61]. As there is data that demonstrating that thiol-based antioxidants exert beneficial effects on AT cells and in some cases, this increased intracellular GSH, a possible treatment for AT could be a combination of a low daily dose of pioglitazone combined with N-acetyl-DL-leucine [20,54,55,56,57]. The latter drug was demonstrated to exert beneficial effects on an individual with AT [57]. Based on present data, such a combination treatment may benefit individuals with AT by simultaneously providing an increased source of reduced antioxidant thiol compounds and pharmacologic agent that increases reduced glutathione in AT cells. Assessing and testing the potential of pioglitazone as possible treatment for AT could be addressed in several ways.

### 3.1. A Clinical Trial Examining the Possible Beneficial Effects of Pioglitazone on AT Patients

To ascertain if pioglitazone has clinical utility on the treatment of AT a clinical trial on AT children with and without pioglitazone treatment would be needed. Parameters such as the SARA score, combined with analyses of the children’s quality of life, blood chemistries with examination of the reduced to oxidized GSH ratio, and possibly parameters related to insulins resistance, such as insulin sensitivity, plasma non-esterified fatty acids concentrations, and fasting respiratory quotient could be examined [19,43,44,45]. Additionally, urine cytology could be employed to look for any possible pioglitazone-induced urothelial atypia or dysplasia. This assessment could be either done at the conclusion of the clinical trial or during its course. N-acetyl-DL-leucine could be added to this clinical trial to access the possible beneficial effects of this antioxidant with pioglitazone, although this would complicate the clinical trial. The previous study finding that pioglitazone may be an appropriate first line treatment for diabetes in AT patients indicates that such a trial may be useful [61]. Additionally, a clinical trial examining the value of metformin and pioglitazone in treating AT is being conducted in the United Kingdom and may give data comparing the possible beneficial effect of pioglitazone and/or metformin on AT (ClinicalTrials.gov ID NCT02733679).

### 3.2. Administering Pioglitazone to Atm-Deficient Mice

Administering pioglitazone to *Atm*-deficient mice would allow the assessment of the drugs effect on parameters such as GSH redox status, mitochondrial function, motor skills and coordination, DNA stability, and bladder cancer risk in an AT animal model. Importantly, the study could be done for a relatively long time and upon study completion, the bladder and genital-urinary tracts of the *Atm*-deficient mice could be histologically examined for pioglitazone-induced urothelial atypia.

### 3.3. Testing This Hypothesis on Cell Lines

An easy initial way to test this hypothesis would be to test the ideas presented here on cell lines. While pioglitazone exerts beneficial effects on AT cells in culture, it would be useful to see if the drug alters AMPK-α phosphorylation in AT cells. Additionally, LKB1 or another kinase that phosphorylates AMPK-α should show be dysregulated in AT. Also, the alterations in signaling pathways downstream of AMPK-α should be both dysregulated in AT and altered by pioglitazone treatment. These experiments would be relatively easy to perform.

## Figures and Tables

**Figure 1 biomolecules-14-01264-f001:**
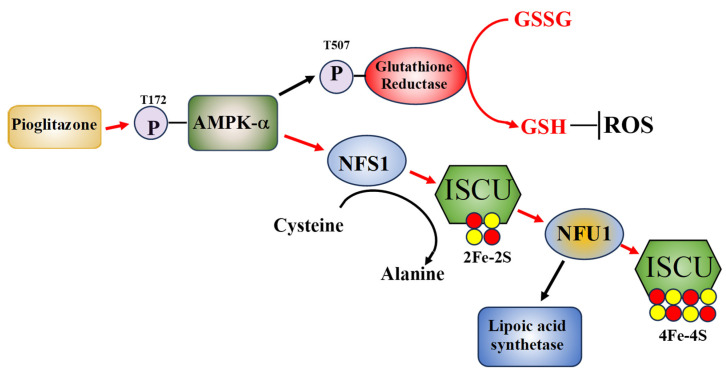
A hypothetical model of how pioglitazone may improve AT cell function. Pioglitazone exposure initiates AMPK-a phosphorylation which in turn activates glutathione reductase through T507 phosphorylation, increasing reduced GSH, lowering cellular ROS levels. Additionally, phosphorylated AMPK-a stabilizes the ISCU complex and increases NFS1 and NFU1 activities and NFS1 mRNA expression levels, with NFU1 activity increasing lipoic acid synthase activity. The pathways that pioglitazone is known to effect are shown with red arrows [34].

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
