# Peer review of "Pioglitazone as a Possible Treatment for Ataxia-Telangiectasia"

_biomolecules, 2024, doi:10.3390/biom14101264_

Round 1

Reviewer 1 Report

Comments and Suggestions for Authors

With the present paper the Author wants to support the hypothesis of a profitable role of pioglitazone in the treatment of patients suffering Ataxia Telangiectasia (AT). He collected data from the literature on the role of pioglitazone as antioxidant and antidiabetic, both in experimental animals as well in human suffering diabetes. The controversial role of the drug in producing bladder cancer is discussed.

The use of antioxidant in AT patients is a frequent therapeutic approach; nevertheless no long-term benificial effects have been demonstrated but generic.

I see two major points againts the treatment with pioglitazone: a) the reported recommendation to not use this drug in pediatric patients, as the majority of AT are; b) the possible increased risk of bladder cancer in individuals, like AT, with a high proneness to develop malignancies. Moreover, bladder is reported as a site of telangiectasias in AT patients.

On my opinion, the Author must stress the need of a trial before recommending pioglitazione to AT patients, also those suffering diabetes.

Based on these observations I suggest to change the title of the paper as following:

Pioglitazone as a possible treatment for Ataxia Telangiectasia

Comments on the Quality of English Language

Lane 8-9: there is a redundancv between high oxidant sensitivity and impaired antioxidant capacities

Lane 21-54-68-83: ATM referred to the gene must be written in italics; ATM refers to the protein

Lane 38: variable instead of viable

Lane 43: ‘individuals with T patients’ is redundant; werite AT patients or AT individuals

Lane 51 : a-fetoprotein must be written alfa-fetoprotein in the absence of the greek letter

Lane 64 : insert a comma between recombination and DNA repair

Lane 79 : seen, not seem

Lane 279 : telangiectasia, not telangeictasia

Finally, being the disease called Ataxia Telangiectasia, I belive the correct acronymous is AT, nor A-T

Author Response

Reveiwer 1:

With the present paper the Author wants to support the hypothesis of a profitable role of pioglitazone in the treatment of patients suffering Ataxia Telangiectasia (AT). He collected data from the literature on the role of pioglitazone as antioxidant and antidiabetic, both in experimental animals as well in human suffering diabetes. The controversial role of the drug in producing bladder cancer is discussed.

The use of antioxidant in AT patients is a frequent therapeutic approach; nevertheless no long-term benificial effects have been demonstrated but generic.

I see two major points againts the treatment with pioglitazone: a) the reported recommendation to not use this drug in pediatric patients, as the majority of AT are; b) the possible increased risk of bladder cancer in individuals, like AT, with a high proneness to develop malignancies. Moreover, bladder is reported as a site of telangiectasias in AT patients.

On my opinion, the Author must stress the need of a trial before recommending pioglitazione to AT patients, also those suffering diabetes.

Based on these observations I suggest to change the title of the paper as following:

Pioglitazone as a possible treatment for Ataxia Telangiectasia

Comments on the Quality of English Language

Lane 8-9: there is a redundancv between high oxidant sensitivity and impaired antioxidant capacities

Lane 21-54-68-83: ATM referred to the gene must be written in italics; ATM refers to the protein

Lane 38: variable instead of viable

Lane 43: ‘individuals with T patients’ is redundant; werite AT patients or AT individuals

Lane 51 : a-fetoprotein must be written alfa-fetoprotein in the absence of the greek letter

Lane 64 : insert a comma between recombination and DNA repair

Lane 79 : seen, not seem

Lane 279 : telangiectasia, not telangeictasia

Finally, being the disease called Ataxia Telangiectasia, I belive the correct acronymous is AT, nor A-T

Author’s Answers:

  1. On my opinion, the Author must stress the need of a trial before recommending pioglitazione to AT patients, also those suffering diabetes.

  • This is an important point and was added to the paper.

  1. Based on these observations I suggest to change the title of the paper as following:

«Pioglitazone as a Possible Treatment for Ataxia Telangiectasia»  

  • I concur and this title much improves the paper.

These errors were corrected and changed:

Lane 8-9: there is a redundancv between high oxidant sensitivity and impaired antioxidant capacities

  • I concur, and this was corrected, with “impaired antioxidant capacities, including” removed form lines 8-9.

Lane 21-54-68-83: ATM referred to the gene must be written in italics; ATM refers to the protein

  • I concur, and this was corrected. Where “ATM refers to the protein, italics were not used.

Lane 38: variable instead of viable

  • This error was corrected.

Lane 43: ‘individuals with T patients’ is redundant; werite AT patients or AT individuals

  • This was corrected to “AT patients”
  •  

Lane 51 : a-fetoprotein must be written alfa-fetoprotein in the absence of the greek letter

  • This was corrected to “a-fetoprotein” with a Greek letter.

Lane 64 : insert a comma between recombination and DNA repair       

  • This was corrected.

Lane 79 : seen, not seem

  • This was corrected.

Lane 279 : telangiectasia, not telangeictasia

  • This was corrected.

Finally, being the disease called Ataxia Telangiectasia, I belive the correct acronymous is AT, nor A-T

  • I concur, and A-T was corrected to AT.

Many other changes were made to the text for further clarification and to answer other reviewer’s corrections.  More data was added on AMPK-a phospshorylation in AT and the ideas presented were explored in more detail. Run-on sentences were shortened. 

Reveiwer 2:

This review by Shackelford focus on the discussion of pioglitazone, which is a drug used for type 2 diabetes, may offer therapeutic benefits for A-T by potentially improving some molecular defects associated with the condition. While the review provides some interesting angles, there are still some confusion parts need more works.

In the introduction, the authors listed some potential symptoms caused by ATM gene mutation. However, this list provides limited information regarding how the mutation contribute to the disease process. How this mutation causes the neurological signs? Could this be related to change the neuronal activities? Circuitry level?  etc.  this also applies to the cancer risk. Are there literatures suggesting the process?  

  • This is a very good suggestion. Many of the pathologies in AT are poorly understood, especially the causes of the neurodegeneration. However, this paper address one on the main causes, which are deficits in sulfur metabolism and impaired reduced GSH synthesis. Similarly, the increased cancer risk is due to the very high genomic instability seen in the syndrome. These were added to the paper; however, a very careful review would make to paper much, much longer.

Regarding the ATM and A-T section, what’s the detailed function of normal ATM? While the author pointed out there are more than 1400 ATM mutations, the details about those mutations are missing. Also, how those mutations are triggered in detail?

  • This is a very interesting point. However, most mutations result in complete loss of the ATM protein. The remaining are much less common mutations are not well characterized. The normal functions of ATM are listed in the introduction.

In the section of A-T and dysregulated iron metabolism, while the author points out the mutation could cause any changes in iron level, but the details and/or mechanism of those changes are missing.

  • More was added to the manuscript concerning ATM and dysregulated iron metabolism. Unfortunately, there are only a handful of papers on AT and dysregulated iron metabolism. All but one of these papers are sighted in the manuscript. Dysregulated iron metabolism in AT is an entirely new area.

 How the pioglitazone changes the AMPK-a T172 phosphorylation? Were these happened in specific tissues? This might relate back to the initial symptoms.

  • A discussion of this was added to the text along more data on AMPK-a kinases and how AMPK-a phosphorylation might be altered in AT.

How the increased GSH contribute to the symptoms discussed in the introduction? This needs more work.

  • A discussion of this was added to the manuscript is several places.

Many other changes were made to the text for further clarification and to answer other reviewer’s corrections.  More data was added on AMPK-a phospshorylation in AT and the ideas presented were explored in more detail. Run-on sentences were shortened. 

Reveiwer 3:

This manuscript introduces the therapeutic potential of pioglitazone for ataxia-telangiectasia (A-T). The effects of pioglitazone on A-T have been investigated mainly in A-T model cells. However, the author describes the possibility of the treatment for patients with A-T. This manuscript is well-written and has an impact for the researchers and neurologists regarding A-T. However, several major and minor points should be revised for the publication.

Major points

  1. Regarding the mechanism of pioglitazone, the activation of AMPK is important. Are there any reports how pioglitazone activates AMPK? What kinds of kinase are involved in pioglitazone-triggered phosphorylation of T172 in AMPK-alpha?

- This was added to the text. Exactly what kinase(s) pioglitazone activates in unknown, although it’s likely LKB1. A detailed discussion of the kinases that phosphorylate and activate AMPK-a was added to the manuscription, including how ATM gene loss my impinge on AMPK-a phosphorylation.

  1. Metformin is more famous as an AMPK activator than pioglitazone. I think there are several reports indicating the effects of metformin on A-T cells. The author should also describe the therapeutic potential of metformin against A-T and claim the superiority of pioglitazone for the treatment for A-T, compared with metformin.

-  Data on metformin was added to the text. Overall, metformin does not appear to activate AMPK-a at clinically relevant doses. The on-going clinical trial in the UK using pioglitazone and metformin to treat AT children was added to the manuscript. Also, there are few to no published papers on the effects of metformin on AT cultured cell or AT patients. I consulted to A-T Children’s Project on this subject.

- In my work I found that metformin did not increase Fe-S cluster formation in AT cells. This is left out of the manuscript, as this data was not published. 

- I have been in contact with the parent of an AT child who told me that pioglitazone helped with the child’s NASH. This was not added to the manuscript as it is antidotal.

Minor points are listed in the attached PDF files. 

Minor points

  1. Several sentences are too long to hardly understand. For example, “Cells from individuals~lowered catalase activity [1-5,15-24]” (lines 70-78).

  • This was corrected.

  1. ”AMPK-a” should be replaced with “AMPK-α”. Similarly, “-g” of “peroxisome proliferator activated receptor-g” should be replaced with “-γ”.

  • These changes were made.

  1. The sentences ” When an A-T cell line and ~ DNA beak formation [34].“ (lines 119-126) were hard to understand. The first sentence is replaced with “Pioglitazone; 1) significantly increased ~ DNA break formation in A-T cell line [34].” The second sentence before comma should be replaced with “These effects of pioglitazone were not seen in ATM-overexpressed A-T cell line.” The phrases after “, were pioglitazone~” (line 124) were not necessary and could be deleted.

  • This part of the manuscript was extensively re-written and data was added that other reviewers requested.

  1. After the sentence “The reported blood concentrations of pioglitazone are 3-6 mM [38,39].” (lines 136-137), the authors should add the description summarizing this paragraph.

  • This was done and the paragraph was extensively re-written.

  1. Line 164 “a recent study were 8 nondiabetic” should be replaced with “a recent study where 8 nondiabetic” or “a recent study in which 8 nondiabetic”.

  • This was corrected. Other typos were also corrected throughout the manuscript.

  1. The sentences “Multiple parameters ~ fasting respiratory quotient” (lines 166-168) were not necessary and could be deleted.

- This was removed. I did not add to the paper.

  1. The sentence “Among these compounds ~ the frequency of DNA deletions [20,39-44].” (lines 178 182) should be rearranged. Although I did not check all their references, I suggest the modified sentence indicated below. “Among these compounds, a-lipoic acid, N-acetyl cysteine, and N-acetyl-DL-leucine have been shown to increase cellular GSH, increase cell viability, and improve mitochondrial function in A-T cells, and to increase longevity and reduce 8-OH deoxyguanosine levels and the frequency of DNA deletions in Atm-deficient 180 mice [20,39-44].”

  • This part of the manuscript was re-written. More was added, at the requested of another reviewer who wanted the details of how thiol-based antioxidant effect AT cells.

Many other changes were made to the text for further clarification and to answer other reviewer’s corrections.  More data was added on AMPK-a phospshorylation in AT and the ideas presented were explored in more detail. Numerous run-on sentences were shortened. 

Reviewer 2 Report

Comments and Suggestions for Authors

This review by Shackelford focus on the discussion of pioglitazone, which is a drug used for type 2 diabetes, may offer therapeutic benefits for A-T by potentially improving some molecular defects associated with the condition. While the review provides some interesting angles, there are still some confusion parts need more works.

In the introduction, the authors listed some potential symptoms caused by ATM gene mutation. However, this list provides limited information regarding how the mutation contribute to the disease process. How this mutation causes the neurological signs? Could this be related to change the neuronal activities? Circuitry level?  etc.  this also applies to the cancer risk. Are there literatures suggesting the process?  

Regarding the ATM and A-T section, what’s the detailed function of normal ATM? While the author pointed out there are more than 1400 ATM mutations, the details about those mutations are missing. Also, how those mutations are triggered in detail?

In the section of A-T and dysregulated iron metabolism, while the author points out the mutation could cause any changes in iron level, but the details and/or mechanism of those changes are missing.

 How the pioglitazone changes the AMPK-a T172 phosphorylation? Were these happened in specific tissues? This might relate back to the initial symptoms.

How the increased GSH contribute to the symptoms discussed in the introduction? This needs more work.

Comments on the Quality of English Language

NA

Author Response

Reveiwer 2:

This review by Shackelford focus on the discussion of pioglitazone, which is a drug used for type 2 diabetes, may offer therapeutic benefits for A-T by potentially improving some molecular defects associated with the condition. While the review provides some interesting angles, there are still some confusion parts need more works.

In the introduction, the authors listed some potential symptoms caused by ATM gene mutation. However, this list provides limited information regarding how the mutation contribute to the disease process. How this mutation causes the neurological signs? Could this be related to change the neuronal activities? Circuitry level?  etc.  this also applies to the cancer risk. Are there literatures suggesting the process?  

  • This is a very good suggestion. Many of the pathologies in AT are poorly understood, especially the causes of the neurodegeneration. However, this paper address one on the main causes, which are deficits in sulfur metabolism and impaired reduced GSH synthesis. Similarly, the increased cancer risk is due to the very high genomic instability seen in the syndrome. These were added to the paper; however, a very careful review would make to paper much, much longer.

Regarding the ATM and A-T section, what’s the detailed function of normal ATM? While the author pointed out there are more than 1400 ATM mutations, the details about those mutations are missing. Also, how those mutations are triggered in detail?

  • This is a very interesting point. However, most mutations result in complete loss of the ATM protein. The remaining are much less common mutations are not well characterized. The normal functions of ATM are listed in the introduction.

In the section of A-T and dysregulated iron metabolism, while the author points out the mutation could cause any changes in iron level, but the details and/or mechanism of those changes are missing.

  • More was added to the manuscript concerning ATM and dysregulated iron metabolism. Unfortunately, there are only a handful of papers on AT and dysregulated iron metabolism. All but one of these papers are sighted in the manuscript. Dysregulated iron metabolism in AT is an entirely new area.

 How the pioglitazone changes the AMPK-a T172 phosphorylation? Were these happened in specific tissues? This might relate back to the initial symptoms.

  • A discussion of this was added to the text along more data on AMPK-a kinases and how AMPK-a phosphorylation might be altered in AT.

How the increased GSH contribute to the symptoms discussed in the introduction? This needs more work.

  • A discussion of this was added to the manuscript is several places.

Many other changes were made to the text for further clarification and to answer other reviewer’s corrections.  More data was added on AMPK-a phospshorylation in AT and the ideas presented were explored in more detail. Run-on sentences were shortened. 

Reviewer 3 Report

Comments and Suggestions for Authors

This manuscript introduces  the therapeutic potential of pioglitazone for  ataxia-telangiectasia (A-T). The effects of pioglitazone on A-T has been investigated mainly in A-T model cells. However, the author describe the possibility of the treatment for patients with A-T. This manuscript is well-written and has an impact for the researchers and neurologists regarding A-T. However, several major and minor points should be revised for the publication.

Major points

1. Regarding the mechanism of pioglitazone, the activaiton of AMPK is important. Are there any reports how pioglitazone activates AMPK? What kinds of kinase are involved in pioglitazone-triggered phosphorylation of T172 in AMPK-alpha?

2. Metfromin is more famous as an AMPK activator than pioglitazone. I think there are several reports indicating the effects of metfromin on A-T cells. The author should also describe the therapeutic potential of metformin against A-T and claim the superiorioty of pioglitazone for the tretment for A-T, compared with metformin.

Minor points are listed in the attached PDF files. 

Comments on the Quality of English Language

The problems of English are listed in the minor points (attached files). One of the problems is several sentences are too long to hardly understand. 

Author Response

Reveiwer 3:

This manuscript introduces the therapeutic potential of pioglitazone for ataxia-telangiectasia (A-T). The effects of pioglitazone on A-T have been investigated mainly in A-T model cells. However, the author describes the possibility of the treatment for patients with A-T. This manuscript is well-written and has an impact for the researchers and neurologists regarding A-T. However, several major and minor points should be revised for the publication.

Major points

  1. Regarding the mechanism of pioglitazone, the activation of AMPK is important. Are there any reports how pioglitazone activates AMPK? What kinds of kinase are involved in pioglitazone-triggered phosphorylation of T172 in AMPK-alpha?

- This was added to the text. Exactly what kinase(s) pioglitazone activates in unknown, although it’s likely LKB1. A detailed discussion of the kinases that phosphorylate and activate AMPK-a was added to the manuscription, including how ATM gene loss my impinge on AMPK-a phosphorylation.

  1. Metformin is more famous as an AMPK activator than pioglitazone. I think there are several reports indicating the effects of metformin on A-T cells. The author should also describe the therapeutic potential of metformin against A-T and claim the superiority of pioglitazone for the treatment for A-T, compared with metformin.

-  Data on metformin was added to the text. Overall, metformin does not appear to activate AMPK-a at clinically relevant doses. The on-going clinical trial in the UK using pioglitazone and metformin to treat AT children was added to the manuscript. Also, there are few to no published papers on the effects of metformin on AT cultured cell or AT patients. I consulted to A-T Children’s Project on this subject.

- In my work I found that metformin did not increase Fe-S cluster formation in AT cells. This is left out of the manuscript, as this data was not published. 

- I have been in contact with the parent of an AT child who told me that pioglitazone helped with the child’s NASH. This was not added to the manuscript as it is antidotal.

Minor points are listed in the attached PDF files. 

Minor points

  1. Several sentences are too long to hardly understand. For example, “Cells from individuals~lowered catalase activity [1-5,15-24]” (lines 70-78).

  • This was corrected.

  1. ”AMPK-a” should be replaced with “AMPK-α”. Similarly, “-g” of “peroxisome proliferator activated receptor-g” should be replaced with “-γ”.

  • These changes were made.

  1. The sentences ” When an A-T cell line and ~ DNA beak formation [34].“ (lines 119-126) were hard to understand. The first sentence is replaced with “Pioglitazone; 1) significantly increased ~ DNA break formation in A-T cell line [34].” The second sentence before comma should be replaced with “These effects of pioglitazone were not seen in ATM-overexpressed A-T cell line.” The phrases after “, were pioglitazone~” (line 124) were not necessary and could be deleted.

  • This part of the manuscript was extensively re-written and data was added that other reviewers requested.

  1. After the sentence “The reported blood concentrations of pioglitazone are 3-6 mM [38,39].” (lines 136-137), the authors should add the description summarizing this paragraph.

  • This was done and the paragraph was extensively re-written.

  1. Line 164 “a recent study were 8 nondiabetic” should be replaced with “a recent study where 8 nondiabetic” or “a recent study in which 8 nondiabetic”.

  • This was corrected. Other typos were also corrected throughout the manuscript.

  1. The sentences “Multiple parameters ~ fasting respiratory quotient” (lines 166-168) were not necessary and could be deleted.

- This was removed. I did not add to the paper.

  1. The sentence “Among these compounds ~ the frequency of DNA deletions [20,39-44].” (lines 178 182) should be rearranged. Although I did not check all their references, I suggest the modified sentence indicated below. “Among these compounds, a-lipoic acid, N-acetyl cysteine, and N-acetyl-DL-leucine have been shown to increase cellular GSH, increase cell viability, and improve mitochondrial function in A-T cells, and to increase longevity and reduce 8-OH deoxyguanosine levels and the frequency of DNA deletions in Atm-deficient 180 mice [20,39-44].”

  • This part of the manuscript was re-written. More was added, at the requested of another reviewer who wanted the details of how thiol-based antioxidant effect AT cells.

Many other changes were made to the text for further clarification and to answer other reviewer’s corrections.  More data was added on AMPK-a phospshorylation in AT and the ideas presented were explored in more detail. Numerous run-on sentences were shortened. 

Round 2

Reviewer 2 Report

Comments and Suggestions for Authors

the authors addressed my concerns. 

Author Response

Reviewer 2:

All the comments were answered, and all the changes made to the manuscript are now highlighted in red.              Please note that the change made were extensive and 16 additional references were added to the paper. Therefore, a lot of the manuscript in now red, to designate where changes were made.

Reveiwer 2 Comments:

This review by Shackelford focus on the discussion of pioglitazone, which is a drug used for type 2 diabetes, may offer therapeutic benefits for A-T by potentially improving some molecular defects associated with the condition. While the review provides some interesting angles, there are still some confusion parts need more works.

In the introduction, the authors listed some potential symptoms caused by ATM gene mutation. However, this list provides limited information regarding how the mutation contribute to the disease process. How this mutation causes the neurological signs? Could this be related to change the neuronal activities? Circuitry level?  etc.  this also applies to the cancer risk. Are there literatures suggesting the process?  

  • This is a very good suggestion. Many of the pathologies in AT are poorly understood, especially the causes of the neurodegeneration. However, this paper address one on the main causes, which are deficits in sulfur metabolism and impaired reduced GSH synthesis. Similarly, the increased cancer risk is due to the very high genomic instability seen in the syndrome. These were added to the paper; however, a very careful review would make to paper much, much longer.

Regarding the ATM and A-T section, what’s the detailed function of normal ATM? While the author pointed out there are more than 1400 ATM mutations, the details about those mutations are missing. Also, how those mutations are triggered in detail?

  • This is a very interesting point. However, most mutations result in complete loss of the ATM protein. The remaining are much less common mutations are not well characterized. The normal functions of ATM are listed in the introduction.

In the section of A-T and dysregulated iron metabolism, while the author points out the mutation could cause any changes in iron level, but the details and/or mechanism of those changes are missing.

  • More was added to the manuscript concerning ATM and dysregulated iron metabolism. Unfortunately, there are only a handful of papers on AT and dysregulated iron metabolism. All but one of these papers are sighted in the manuscript. Dysregulated iron metabolism in AT is an entirely new area.

 How the pioglitazone changes the AMPK-a T172 phosphorylation? Were these happened in specific tissues? This might relate back to the initial symptoms.

  • A discussion of this was added to the text along more data on AMPK-a kinases and how AMPK-a phosphorylation might be altered in AT.

How the increased GSH contribute to the symptoms discussed in the introduction? This needs more work.

  • A discussion of this was added to the manuscript is several places.

Many other changes were made to the text for further clarification and to answer other reviewer’s corrections.  More data was added on AMPK-a phospshorylation in AT and the ideas presented were explored in more detail. Run-on sentences were shortened. 

Reviewer 3 Report

Comments and Suggestions for Authors

Mostly, I satisfied the authors' revisions. However, several points are left to be revised.

1. Full spell of LKB1 is necessary.

2. Minor point 2 remains unmodified. Most "AMPK-a" are still "AMPK-a". Two "peroxisome proliferator-267 activated receptor-g" are not corrected.

Comments on the Quality of English Language

Quality fo English language is quite improved in the revised manuscript.

Author Response

Reviewer 3:

All the comments were answered, and all the changes made to the manuscript are now highlighted in red.              Please note that the change made were extensive and 16 additional references were added to the paper. Therefore, a lot of the manuscript in now red, to designate where changes were made.

Reveiwer 3 Comments:

This manuscript introduces the therapeutic potential of pioglitazone for ataxia-telangiectasia (A-T). The effects of pioglitazone on A-T have been investigated mainly in A-T model cells. However, the author describes the possibility of the treatment for patients with A-T. This manuscript is well-written and has an impact for the researchers and neurologists regarding A-T. However, several major and minor points should be revised for the publication.

Major points

  1. Regarding the mechanism of pioglitazone, the activation of AMPK is important. Are there any reports how pioglitazone activates AMPK? What kinds of kinase are involved in pioglitazone-triggered phosphorylation of T172 in AMPK-alpha?

- This was added to the text. Exactly what kinase(s) pioglitazone activates in unknown, although it’s likely LKB1. A detailed discussion of the kinases that phosphorylate and activate AMPK-a was added to the manuscription, including how ATM gene loss my impinge on AMPK-a phosphorylation.

  1. Metformin is more famous as an AMPK activator than pioglitazone. I think there are several reports indicating the effects of metformin on A-T cells. The author should also describe the therapeutic potential of metformin against A-T and claim the superiority of pioglitazone for the treatment for A-T, compared with metformin.

-  Data on metformin was added to the text. Overall, metformin does not appear to activate AMPK-a at clinically relevant doses. The on-going clinical trial in the UK using pioglitazone and metformin to treat AT children was added to the manuscript. Also, there are few to no published papers on the effects of metformin on AT cultured cell or AT patients. I consulted to A-T Children’s Project on this subject.

- In my work I found that metformin did not increase Fe-S cluster formation in AT cells. This is left out of the manuscript, as this data was not published. 

- I have been in contact with the parent of an AT child who told me that pioglitazone helped with the child’s NASH. This was not added to the manuscript as it is antidotal.

Minor points are listed in the attached PDF files. 

Minor points

  1. Several sentences are too long to hardly understand. For example, “Cells from individuals~lowered catalase activity [1-5,15-24]” (lines 70-78).

  • This was corrected.

  1. ”AMPK-a” should be replaced with “AMPK-α”. Similarly, “-g” of “peroxisome proliferator activated receptor-g” should be replaced with “-γ”.

  • These changes were made.

  1. The sentences ” When an A-T cell line and ~ DNA beak formation [34].“ (lines 119-126) were hard to understand. The first sentence is replaced with “Pioglitazone; 1) significantly increased ~ DNA break formation in A-T cell line [34].” The second sentence before comma should be replaced with “These effects of pioglitazone were not seen in ATM-overexpressed A-T cell line.” The phrases after “, were pioglitazone~” (line 124) were not necessary and could be deleted.

  • This part of the manuscript was extensively re-written and data was added that other reviewers requested.

  1. After the sentence “The reported blood concentrations of pioglitazone are 3-6 mM [38,39].” (lines 136-137), the authors should add the description summarizing this paragraph.

  • This was done and the paragraph was extensively re-written.

  1. Line 164 “a recent study were 8 nondiabetic” should be replaced with “a recent study where 8 nondiabetic” or “a recent study in which 8 nondiabetic”.

  • This was corrected. Other typos were also corrected throughout the manuscript.

  1. The sentences “Multiple parameters ~ fasting respiratory quotient” (lines 166-168) were not necessary and could be deleted.

- This was removed. I did not add to the paper.

  1. The sentence “Among these compounds ~ the frequency of DNA deletions [20,39-44].” (lines 178 182) should be rearranged. Although I did not check all their references, I suggest the modified sentence indicated below. “Among these compounds, a-lipoic acid, N-acetyl cysteine, and N-acetyl-DL-leucine have been shown to increase cellular GSH, increase cell viability, and improve mitochondrial function in A-T cells, and to increase longevity and reduce 8-OH deoxyguanosine levels and the frequency of DNA deletions in Atm-deficient 180 mice [20,39-44].”

  • This part of the manuscript was re-written. More was added, at the requested of another reviewer who wanted the details of how thiol-based antioxidant effect AT cells.

Many other changes were made to the text for further clarification and to answer other reviewer’s corrections.  More data was added on AMPK-a phospshorylation in AT and the ideas presented were explored in more detail. Numerous run-on sentences were shortened. 
